# Statistical Optimisation of Diesel Biodegradation at Low Temperatures by an Antarctic Marine Bacterial Consortium Isolated from Non-Contaminated Seawater

**DOI:** 10.3390/microorganisms9061213

**Published:** 2021-06-03

**Authors:** Nur Nadhirah Zakaria, Claudio Gomez-Fuentes, Khalilah Abdul Khalil, Peter Convey, Ahmad Fareez Ahmad Roslee, Azham Zulkharnain, Suriana Sabri, Noor Azmi Shaharuddin, Leyla Cárdenas, Siti Aqlima Ahmad

**Affiliations:** 1Department of Biochemistry, Faculty of Biotechnology and Biomolecular Sciences, Universiti Putra Malaysia, Serdang 43400, Selangor, Malaysia; nadhirahairakaz@gmail.com (N.N.Z.); fareezlee@yahoo.com (A.F.A.R.); noorazmi@upm.edu.my (N.A.S.); 2Department of Chemical Engineering, Universidad de Magallanes, Avda. Bulnes, Punta Arenas 01855, Región de Magallanes y Antártica Chilena, Chile; claudio.gomez@umag.cl; 3Center for Research and Antarctic Environmental Monitoring (CIMAA), Universidad de Magallanes, Avda. Bulnes, Punta Arenas 01855, Región de Magallanes y Antártica Chilena, Chile; 4School of Biology, Faculty of Applied Sciences, Universiti Teknologi MARA, Shah Alam 40450, Selangor, Malaysia; khali552@uitm.edu.my; 5British Antarctic Survey, NERC, High Cross, Madingley Road, Cambridge CB3 0ET, UK; pcon@bas.ac.uk; 6Department of Zoology, University of Johannesburg, PO Box 524, Auckland Park 2006, South Africa; 7Department of Bioscience and Engineering, College of Systems Engineering and Science, Shibaura Institute of Technology, 307 Fukasaku, Minumaku, Saitama 337-8570, Japan; azham@shibaura-it.ac.jp; 8Department of Microbiology, Faculty of Biotechnology and Biomolecular Sciences, Universiti Putra Malaysia, Serdang 43400, Selangor, Malaysia; suriana@upm.edu.my; 9Centro Fondap Ideal, Insituto de Ciencias Ambientales y Evolutivas, Facultad de Ciencias, Universidad Austral de Chile, Casilla 567, Valdivia, Chile; leylacardenas@uach.cl; 10National Antarctic Research Centre, B303 Level 3, Block B, IPS Building, Universiti Malaya, Kuala Lumpur 50603, Malaysia

**Keywords:** Antarctica, biodegradation, diesel, microbial consortium, seawater

## Abstract

Hydrocarbon pollution is widespread around the globe and, even in the remoteness of Antarctica, the impacts of hydrocarbons from anthropogenic sources are still apparent. Antarctica’s chronically cold temperatures and other extreme environmental conditions reduce the rates of biological processes, including the biodegradation of pollutants. However, the native Antarctic microbial diversity provides a reservoir of cold-adapted microorganisms, some of which have the potential for biodegradation. This study evaluated the diesel hydrocarbon-degrading ability of a psychrotolerant marine bacterial consortium obtained from the coast of the north-west Antarctic Peninsula. The consortium’s growth conditions were optimised using one-factor-at-a-time (OFAT) and statistical response surface methodology (RSM), which identified optimal growth conditions of pH 8.0, 10 °C, 25 ppt NaCl and 1.5 g/L NH_4_NO_3_. The predicted model was highly significant and confirmed that the parameters’ salinity, temperature, nitrogen concentration and initial diesel concentration significantly influenced diesel biodegradation. Using the optimised values generated by RSM, a mass reduction of 12.23 mg/mL from the initial 30.518 mg/mL (4% (*w/v*)) concentration of diesel was achieved within a 6 d incubation period. This study provides further evidence for the presence of native hydrocarbon-degrading bacteria in non-contaminated Antarctic seawater.

## 1. Introduction

Antarctica is administered under the Antarctic Treaty, which came into force in 1961 and currently includes 54 member states (known as Parties), 29 of which are full ‘Consultative Parties’, which participate in consensus decision making at the annual Antarctic Treaty Consultative Meetings. During the Treaty’s existence, various environmental regulations have been adopted and enforced to protect and preserve the natural environment of Antarctica. Human activity in Antarctica carries environmental risks associated with using fossil fuels, which are relied upon heavily as the primary energy source for powering research stations, as well as vessels, vehicles and aircraft [1]. Despite this, oil spills have occurred throughout the life of the Treaty, affecting all major types of the Antarctic environment (marine, terrestrial, ice) [2,3,4,5]. Small-scale spills have been reported mostly in the vicinity of research stations, the majority of which lie near the coastline and field camps [6], with (much) larger spills associated with shipping accidents, station refueling and air operations [4,7,8]. Many research stations confirm their reliance on diesel fuel [9,10,11]. Although precise figures are difficult to obtain, in 2004, annual fuel consumption reported by research stations amounted to 90 million litres, 75% of which was diesel fuel [1].

Despite the scale of current and predicted future shipping operations using the ‘northern sea route’ in the Arctic Ocean as multi-year sea ice extent rapidly decreases, the existence of large-scale hydrocarbon extraction industries both on land and in the marine environment of the Arctic (and associated history of very large-scale spills, including major shipping accidents involving supertankers), the repercussions of hydrocarbon pollution in Antarctica are potentially more significant than in the northern polar regions. This is due to Antarctica’s greater isolation, the general lack of large-scale response facilities and logistics and the distances required to deliver needed equipment and personnel to any spill location. The sources of hydrocarbons related to anthropogenic activities in Antarctica include fossil fuel combustion, fuel storage, accidental oil spills and sewage discharge [12,13,14,15]. Shipping accidents and the ship-to-shore refuelling activities necessary for the running of Antarctic programs have proven to be the most hazardous activities, as they can potentially lead to very large spills affecting both marine and terrestrial environments [10,16]. Hydrocarbon pollution is both long-lived and pervasive in Antarctica [17]. Hydrocarbons detected in surface waters of Antarctica have been traced to melt from pack-ice, permafrost and continental ice and snow, originating from atmospheric deposition [18].

The initial impact of hydrocarbons spilled into marine waters occurs near the sea’s surface, rather than in the underlying water column [19]. Obligate hydrocarbonoclastic bacteria (OHCB) are recognised for their ability to utilise hydrocarbons almost exclusively as a sole carbon source, with the added significance of being found in many marine environments [20]. Species related to OHCB have been reported in the pristine waters of Antarctica [21,22,23]. OHCB are regarded as key players in the natural cleansing of oil-polluted marine systems [22]. OHCB species have also been reported to utilise the well-known intermediate metabolites of hydrocarbons, pyruvate and pyruvic acid, as a carbon source [24].

Native microbial consortia that are present in marine environments are likely to degrade the major fractions of hydrocarbon pollutants, ultimately reducing the impact of oil spills [25]. However, the impact of hydrocarbon pollution even at a very local scale in Antarctica is magnified because of the combination of Antarctica’s largely pristine environment and its sensitive and often endemic biological communities [26]. In marine ecosystems, even small-scale oil spills can result in toxic dissolved compounds becoming bioavailable [27]. It has long been recognised that hydrocarbons, including polyaromatic hydrocarbons, are able to support microbial growth [28] and microbial biodegradation is increasingly widely accepted as a primary dissipation mechanism for most organic pollutants [25,29]. However, under the terms of the Protocol for Environmental Protection to the Antarctic Treaty, the introduction of non-native organisms to the natural environment of Antarctica is prohibited, as is the use of many classes of chemicals that might themselves cause damage to the environment. Thus, any application of bioremediation approaches in Antarctica requires the identification and use of native microbial species or consortia.

Microbial remediation technology can make a key contribution to ecological security when dealing with petroleum hydrocarbon-polluted environments, due to its relatively low cost and minimal environmental impact [30]. Technologies that exploit progressively more effective strains of microbes have been developed over time, including some broad-spectrum hydrocarbon-degrading bacteria [31,32]. However, no single bacteria can degrade the entire petroleum hydrocarbon spectrum in isolation. In recent years, considerable effort has been devoted to exploiting marine organisms in the bioremediation of pollutants [33]. Extensive studies of hydrocarbon biodegradation by cold-adapted single strains of marine bacteria have been explored [22,23,34].

Studies have indicated that biodegradation potential could be improved through the application of bacterial consortia (i.e., combinations of multiple species/strains). Such consortia can offer more diverse catabolic genes and their synergistic effects may be beneficial in achieving more efficient mineralisation of pollutants [35]. Consortia have yielded better results in both aliphatic and polyaromatic chain hydrocarbon degradation [36], and have also increasingly addressed the population dynamics of Antarctic marine microbial communities under exposure to hydrocarbons [4,37,38,39]. However, there remains a lack of consortium studies based on the native microbiota of Antarctic seawater, despite some of the largest Antarctic hydrocarbon pollution events taking place in the marine environment [16].

The effectiveness of bioremediation technology applications for petroleum hydrocarbon pollution is increasing [40] and increasing interest is being shown in its use in Antarctica [41,42]. Bioremediation applications need to be designed in a site-specific manner and be based on detailed knowledge of the local native microbial communities and their responses to exposure to contaminants [43].

Studies aimed at enhancing the biodegradation of diesel have used optimisation approaches through multivariate statistical techniques, such as response surface methodology (RSM). RSM includes a series of statistically designed experiments that can effectively minimise error in determining the effects of parameters and their interactions and can be used to determine the optimum conditions required to maximise biodegradation [42,44]. With this background, the current study sets out to assess the ability of a native bacterial consortium obtained from unpolluted Antarctic seawater to effectively biodegrade diesel using RSM.

## 2. Materials and Methods

### 2.1. Sampling and Media

Seawater samples were collected from the coast close to the Chilean research station, General Bernardo O’Higgins on the Trinity Peninsula, north-west Antarctic Peninsula (63°19′15″ S 57°53′55″ W). Samples were collected during 2 successive austral summers in December-January 2017/18 and 2018/19. Water samples (50 mL) were obtained between 0–15 m depth using sterile polycarbonate bottles and were kept at 4 °C until processing. After being rapidly returned to the laboratory, the samples were divided into 2 mL aliquots and immediately frozen at −80 °C until required.

All chemicals used were of analytical grade unless otherwise specified. Petronas diesel, locally obtained from a fuel station in Selangor, Malaysia, was used as the sole carbon source in the screening and optimisation part of this study. The diesel was filtered using a sterile 0.22 µm pore size filter syringe and stored at room temperature (25 °C) in a sterile, amber glass Schott bottle to avoid photooxidation [42].

### 2.2. Screening for Diesel Hydrocarbon Biodegradation

One millilitre of each water sample was precultured into nutrient broth and incubated at 10 °C to obtain the initial bacterial consortia. Precultured samples were incubated until visible cell turbidity appeared (48–144 h) before being centrifuged at 8000× rpm for 10 min, and the resulting pellets were washed once using 1× phosphate-buffered saline (PBS: 137 mM NaCl, 2.7 mM KCl in 10 mM phosphate buffer, pH 7.4). The pellets were re-suspended in PBS to an absorbance reading of OD_600_ = 1.0, which yielded a CFU/mL of 38 × 10^20^. Preparation of bacterial suspension followed this procedure throughout the study except when indicated otherwise. Diesel degrading abilities of the samples were assessed using a standardised Bushnell‒Haas (BH) salt medium [45]. This was adjusted to pH 7.0, further supplemented with 20 g NaCl (*w/v*), equivalent to 20 ppt salinity, and 1.0% (*v/v*) initial diesel fuel concentration added as the sole carbon source. All experiments were performed in triplicate in an overall volume of 50 mL in flasks. Flasks without inoculum served as controls to assess any abiotic degradation of the diesel. Selection of the bacterial consortium to be used in the subsequent parts of the study was based on the combination of degree of diesel degradation achieved and growth rate.

### 2.3. Optimisation of Diesel Degradation Using One-Factor-at-a-Time (OFAT)

Diesel degradation was first studied using the OFAT approach by optimising the growth conditions of the selected Antarctic marine bacterial consortium. The parameters optimised were pH, salinity, temperature, nitrogen source and concentration and initial substrate (diesel) concentration. Each parameter was optimised in 50 mL BH media on an orbital shaker (150 rpm) at 10 °C (the temperature was optimised separately) (Table 1). After a 7 d incubation period, 1 mL from each flask was centrifuged at 13,000× rpm for 10 min to yield an inoculum cell pellet. Growth was measured using suspension turbidity as a proxy, using a UV-Vis spectrophotometer and a wavelength of 600 nm (OD_600_).

Bushnell‒Haas (BH) salt medium was supplemented with diesel at 1.0% (*v/v*) except in the final step of optimising initial substrate concentration. Each parameter was optimised successively in the order listed in Table 1. One-way ANOVA was used to test the influence of each parameter on microbial growth and percentage of diesel degradation followed, where significant, by pairwise post hoc comparisons using Tukey’s test.

### 2.4. Quantification of Diesel Degradation

Residual diesel mass was measured gravimetrically using n-hexane extraction [46]. At the end of the incubation period, the remaining diesel in each triplicate flask was extracted using 20 mL of n-hexane, which was then agitated for 60 min at a room temperature of 25 °C. The extraction yielded two layers and the upper organic layer was removed into a glass Petri dish and the solvent evaporated in a fume hood. The percentage of diesel degradation (%) was calculated using Equation (1) [47]. Abiotic control values were subtracted from the residual mass to calculate biodegradation [48].
(1)$$\%\;Biodegradation=\frac{\left[Mass\;of\;resdidual\;diesel\;abiotic\;control\;\left(g\right)-Mass\;of\;residual\;diesel\;treatment\;\left(g\right)\right]}{Mass\;ofdiesel\;in\;abiotic\;control\;\left(g\right)}\;x100\;$$

### 2.5. Response Surface Methodology

Response Surface Methodology is a combination of statistical and mathematical methods that are used to generate a sequence of design experiments that minimises the number of experimental runs required to achieve the desired results. Importantly, unlike OFAT, this methodology allows for the identification of significant pairwise interactions between the tested parameters influencing diesel degradation. Two experimental designs (DoE) were applied in this study, Placket-Burman design (PBD) and Central Composite Design (CCD). Both DoE approaches were analysed using Design-Expert version 12 (Stat Ease, Inc, Minneapolis, MN, USA). The adequacy of the model terms was assessed using analysis of variance (ANOVA). The significance of each model term, also labelled as a coefficient in the equation, was determined by Fisher’s F test and (ANOVA). Probability values <0.05 are accepted as being significant. Adeq precision measures the signal to noise ratio, and a ratio greater than 4 is desirable. Low coefficient of variation values (CV) support the precision and reliability of the model. R^2^ measures the goodness of fit of the values obtained.

#### 2.5.1. Selection of Significant Variables by Plackett-Burman Design

Generally, a Plackett-Burman (PB) design is used to rapidly screen multiple factors to discover the significant independent parameters [49,50]. PBD is useful for economically detecting large main effects by independent parameters assuming that any factor interactions remain negligible compared with the main effects. The factors previously determined in OFAT were further screened using PB factorial design with a first-order polynomial equation. Each factor was tested at high (+1) and low (−1) levels (Table 2). Diesel reduction was set to be the response variable of the DoE. The incubation period for PBD runs was 7 d, as used in OFAT.

The PBD follows the first-order model below.
(2)$$Y=\beta_{0}+\overset{k}{\underset{i=1}{\sum }}\beta_{i}x_{i}\;$$
where *Y* represents the response variable, *x* are the independent factors that influence *Y*, *β_0_* is the intercept, *k* is the number of involved factors and *β_i_* is the ith linear coefficient.

#### 2.5.2. Central Composite Design

After PBD has identified the significant single factors, RSM applies central composite design (CCD) to further optimise diesel biodegradation. The design model is quadratic and is fitted to characterise the nature of the response surface in the favoured experimental region. Table 3 lists the effects of each parameter on diesel degradation, which were studied at five different levels by combining two factorial points, two axial points and a sole central point (+2, +1, 0, −1, −2). The percentage of diesel degradation was again the response variable. The experimental response was fitted to a second-order polynomial regression model including the significant linear, pairwise and quadratic interaction coefficients to predict the optimal conditions. The quadratic mathematical model used is shown in Equation (3).
(3)$$y=\beta_{0}+\overset{k}{\underset{i=1}{\sum }}\beta_{i}x_{i}+\overset{k}{\underset{i=1}{\sum }}\beta_{\mathrm{ii}}x_{i}^{2}+\overset{k}{\underset{1=i\leq j}{\sum }}\beta_{\mathrm{ij}}x_{i}x_{j\;,\;i\;\neq \;j}$$
where y represents the response variable, *x* are the independent factors that influence Y, *β_0_* is the intercept, k is the number of involved factors, β_i_ is the ith linear coefficient, β_ii_ is the quadratic coefficient, β_ij_ the coefficient of interaction effect when i < j, with i and *j* = 1, 2, 3 and i ≠ j. All experiments were performed in triplicate.

## 3. Results

### 3.1. Screening of Diesel Degrading Consortia

A 7 d incubation period was used for all samples to identify the existence of hydrocarbon degrading ability, with samples shown separately according to the sampling season in Figure 1.

Among the samples tested, the consortium obtained from sample o2b displayed the highest level of diesel degradation (77.97% ± 2.24), although comparable to consortia o1a, o2a and o2c. As consortium o2b showed high degradation abilities with considerably lower growth, it was selected for further study. This sample was obtained close to the coast of Kopaitic Island (Figure 2).

### 3.2. Optimisation of Growth Conditions Using Conventional One-Factor-at-a-Time

#### 3.2.1. Effect of pH, Salinity and Temperature on Diesel Degradation

Analysis of variance (ANOVA) was used to assess the influence of the variables tested on bacterial consortium growth and diesel biodegradation. This identified a significant effect of pH on diesel degradation (F_11,24_ = 4.851, *p* = 0.0006). Tukey’s post hoc tests revealed significant differences between pH 5.8 and 8.0 (*p* = 0.019) and pH 6.5 and 8.0 in the phosphate buffer system (*p* = 0.0002) but no significant differences in the range of pH 7.0–8.5. Degradation achieved at pH 8.0 and 9.0 was significantly different (*p* = 0.0108). Consortium o2b showed maximum growth at pH 7.0 but maximum diesel degradation at pH 7.5 and 8.0 in phosphate buffer, with performance in Tris buffer being slightly better at pH 8.0. The differences in pH also had a significant influence on growth of consortium o2b (F_10,22_ = 15.70, *p* = 0.0003). Tukey’s post hoc comparisons revealed no significant differences between pH 7.0–pH 8.0 for the phosphate buffer. Growth at pH 8.0 in both phosphate and Tris buffer systems was also not significantly different. However, growth at pH 7.5 in phosphate and Tris buffer systems was significantly different (*p* < 0.0029).

ANOVA confirmed that salinity had a significant influence on diesel degradation by bacterial consortium o2b (F_8,10_ = 5.238, *p* = 0.0149). Optimal diesel degradation and bacterial growth were achieved at a salinity of 30 ppt (Figure 3b). Tukey’s post hoc comparisons revealed significant differences in degradation between 0 ppt and 30 ppt (*p* = 0.005) and between 30 and 40 ppt (*p* = 0.0149). Consortium growth generally increased with salinity, with ANOVA confirming a significant influence of salinity on growth F_9,20_ = 21.39, *p* < 0.0001). At 35 ppt and 40 ppt, consortium growth was lower but not significantly different to 30 ppt, as was the degradation of diesel achieved. Tukey’s post hoc comparisons revealed significant differences between 30 ppt and all other lower salinities.

Temperature also significantly influenced diesel degradation (Figure 3c) (F_4,10_ = 78.66, *p* < 0.0001) and microbial growth (F_4,9_ = 19.30, *p* = 0.0002). All post hoc pairwise temperature comparisons on diesel biodegradation were significant (all *p* < 0.0001). Post hoc comparisons revealed no significant influence of temperature on microbial growth between 5 °C and 10 °C, 10 °C and 15 °C and 10 °C and 25 °C. However, higher temperatures, such as 20 °C, led to significant reduction in growth compared to the optimum temperature for growth of 10 °C (*p* = 0.0001).

#### 3.2.2. Effect of Nitrogen Source and Concentration on Diesel Degradation

Figure 4a displays the effects of different nitrogen sources on bacterial growth and degradation of diesel. There were overall significant differences in the influence of the different nitrogen sources on diesel degradation and microbial growth (F_5,12_ = 20.5, *p* < 0.0001). Post hoc tests identified no significant differences in diesel degradation between the nitrogen sources NH_4_NO_3_, NH_4_Cl, NH_4_SO_4_ and urea. A significant difference in consortium growth was identified between NH_4_NO_3_ and urea (*p* = 0.0001). Based on achieving the greatest degradation, ammonium nitrate (NH_4_NO_3_) was selected as the nitrogen source for further optimisation.

The influence of NH_4_NO_3_ concentration on microbial growth and diesel degradation is shown in Figure 4b. Growth and degradation differed significantly between NH_4_NO_3_ concentrations (F_6,14_ = 4.884, *p* = 0.0068 and F_6,14_ = 3.092, *p* = 0.0383, respectively). Maximum degradation was observed at 0.5–1.0 g/L (although not differing significantly up to 2.5 g/L). Tukey’s post hoc comparison revealed that diesel degradation did not differ significantly across all nitrogen concentrations tested. Bacterial growth increased with an increasing concentration of ammonium nitrate. Tukey’s post hoc comparisons revealed no significant differences in the influence of nitrogen concentration on consortium growth between values 0 and 2.0 g/L. Growth was significantly greater at nitrogen concentrations of 2.5 g/L (*p* = 0.0033) and 3.0 g/L (*p* = 0.0197) compared to 0.0 g/L, but did not differ significantly between them.

#### 3.2.3. Effect of Initial Diesel Concentration on Bacterial Consortium Growth and Diesel Degradation

The initial diesel concentration significantly influenced both bacterial consortium growth (F_8,18_ = 17.64, *p* < 0.0001) and diesel degradation (F_8,18_ = 173.7, *p* < 0.0001) (Figure 5). Post hoc comparisons indicated that increasing initial diesel concentration had a significant effect on the absolute amount of diesel that was degraded up to a concentration of 3.5%. However, further increase to 4.0, 4.5 or 5.0% did not lead to any further increase in diesel degradation, and 5.5% led to a considerable decrease in degradation. Bacterial consortium growth generally increased with increasing initial diesel concentration up to 4%.

### 3.3. Application of Response Surface Methodology in the Selection of Significant Variables

#### 3.3.1. Plackett-Burman Design

The PB design matrix was applied to bacterial consortium o2b. Twelve runs were generated (Table 4). Across these runs diesel degradation ranged between 16.58% and 39.35%. The highest value (run 10) was obtained at a temperature of 15 °C, pH 8.0, 25 ppt salinity, 0.5 g/L NH_3_NO_4_ and 1.0% diesel and the lowest (run 4) at 10 °C, pH 7.5, 25 ppt salinity, 2.5 g/L NH_3_NO_4_ and 4.0% diesel. Most other PBD runs at a temperature of 15 °C also gave high diesel biodegradation outcomes (Table 4).

ANOVA identified the parameters that significantly influenced diesel degradation by bacterial consortium o2b, with the only factor not having a significant influence being A (pH), and also confirmed that the overall model was highly significant (Table 5).

#### 3.3.2. Central Composite Design

After identification of the significant parameters in PBD, CCD was implemented to further optimise diesel biodegradation through identification and inclusion of significant pairwise interactions between these parameters. The incubation period used in CCD was 6 d. The results of 30 experimental runs generated by CCD are given in Table 6, with the levels of diesel biodegradation ranging between 17.33% and 48.56%. Run 10 in the model generated a negative value for parameter E (diesel concentration) and was not included in the analysis.

A quadratic model was chosen and ANOVA was used to assess the significance of each model term (Table 7).

To determine the optimal levels of each parameter for maximum diesel oil biodegradation, three-dimensional response surface plots were constructed by plotting the response (diesel degradation, mg/mL) on the z-axis against any two independent parameters, while maintaining other parameters at their optimal values. These surface plots allowed for visualisation of the optimum values of each parameter that yielded the highest response. Figure 6a shows the interaction between salinity and nitrogen concentration. The combination of higher nitrogen concentration and lower salinity led to the greatest diesel degradation, reaching a maximum of 8.1 mg/mL with 1.5 g/L NH_4_NO_3_ and 25 ppt NaCl with temperature and diesel concentration held at 10 °C and 4.0% (*v/v*), respectively. Figure 6b illustrates the interaction between NH_4_NO_3_ concentration and temperature at a constant temperature. Diesel degradation was greatest between temperatures of 11 °C and 13 °C at the highest NH_4_NO_3_ concentration of 1.5 g/L with 8.1 mg/mL mass reduction from 18.54 mg/mL initial diesel concentration. Lower degradation values were predicted for higher temperatures, suggesting the inactivation of hydrocarbon-degrading enzymes and key metabolic players. Figure 6c shows the interaction between temperature and initial diesel concentration. Higher diesel reduction of 11.2 mg/mL was observed at lower temperatures of 10–12 °C.

The model was validated by performing an experimental trial using the predicted optimised conditions displayed in Table 8 (pH 8.0, 25.0 ppt, 1.5 g/L NH_3_NO_4_, 10 °C and 4.0% (*v/v*) diesel). The program predicted a value of 11.66 mg/mL for total diesel mass reduction. These conditions applied experimentally yielded a not significantly different degradation of 12.23 mg/mL ± 1.46 from the initial 30.518 mg/mL (two-tailed t test, *p* = 0.506).

## 4. Discussion

Natural microbial communities can be distinguished into three general groups: saprophytic, symbiotic and parasitic [51]. Hydrocarbon-degrading microorganisms constitute between 1 and 10% of the total number of saprophytic microorganisms in marine microbial communities [52,53], and heterotrophic bacteria near the ocean’s surface decompose 75–95% of the organic matter generated by autotrophic organisms in the photic zone [54]. On this basis, microbial mineralisation of diesel hydrocarbon molecules is primarily expected to occur near the ocean’s surface [55]. The current study relied on bacterial consortium samples obtained from a shallow marine environment with no history or evidence of hydrocarbon contamination. However, even in uncontaminated areas such as these, bacteria capable of hydrocarbon-degradation are still expected to be ubiquitous [22,56]. In the event of hydrocarbon contamination of the natural environment, the autochthonous microbiota initially present are subjected to strong selection pressure, with only those with appropriate resistance characteristics essential for survival and capable of expressing specific enzymatic biodegrading pathways ultimately remaining [57].

### 4.1. Optimisation of Growth Conditions Using Conventional One-Factor-at-a-Time

#### 4.1.1. pH

The pH value of the surrounding environment plays a crucial role in microbial mineralisation of diesel as a sole carbon source. Surface ocean water worldwide, including in the Antarctic, is alkaline, typically ranging from pH 7.9 to 8.2 [58,59]. Generally, the buffering capacity of seawater reaches a minimum at a pH value of approximately 7.5 [60,61], consistent with the non-significant difference in diesel degradation observed here between pH 7.5 and 8.0 in the phosphate buffer system. The pH of the growth medium influences microbial growth and metabolism, enzymatic activities, transport processes and nutrient solubility [62]. Many heterotrophic bacteria, and in particular soil-dwelling species, perform optimally at a pH close to neutral (e.g., [42]). However, marine bacteria can survive and grow in a slightly alkaline environment [63,64]. Marine microbes can also adapt to changes in environmental pH [55], although maintain a cytoplasmic pH between 7.4 and 7.8 [65] that is compatible with optimal function and structural integrity of the cytoplasmic proteins that support growth [66]. The presence of phosphorus-containing compounds in the experimental buffer system used may itself have made a nutritional contribution to the microbial cells encouraging bacterial growth [67]. In the oceanic environment, the natural concentration of P is non-limiting for microbial uptake. However, under oil spill conditions, the concentration of P is altered and it can become a limiting factor [68].

#### 4.1.2. Salinity

The data obtained here are comparable to those of Yakimov et al. [69], who reported optimal bacterial growth between 30–40 ppt salinity. The ocean surrounding the north-western Antarctic Peninsula, where General Bernardo O’Higgins station is located, forms part of the Southern Ocean, which surrounds the entire Antarctic continent and is bounded by the Antarctic Circumpolar, which provides some isolation from the waters of other oceans. The Southern Ocean has a typical salinity range of 33.0 to 34.6 [59]. A study carried out in the western Antarctic Peninsula region, close to the current study’s sampling location, reported a salinity range of 32.5–35.5 ppt in a dataset collected over 19 years [70].

#### 4.1.3. Temperature

The western Antarctic Peninsula (WAP) was one of the most rapidly warming regions on the planet in the second half of the twentieth century, with a surface air temperature warming rate of 3.7 ± 1.6 °C per century [71,72,73]. Surface waters of the Southern Ocean close to the continent have also warmed [74,75], with some evidence also of warming of the Circumpolar Deep Water that surrounds the Antarctic continent [76]. Rodrigues-Blanco et al. [77] suggested that, although temperature and substrate are crucial to bacterial growth, different bacterial species respond differently to temperature. The majority of microbial strains isolated in Antarctic seawater are considered psychrotolerant, rather than true psyhchrophiles [53]. The bacterial consortium o2b studied here is likely to contain a mixture of psychrotolerant and psychrophilic bacteria. Low bacterial growth, such as seen at the higher temperatures used here, can be associated with the denaturation of key cellular components, while low temperatures may restrict bacterial growth through the loss of membrane function [78]. Within the functional temperature range of a given species, microbial metabolism increases with temperature [79], while diesel uptake or assimilation is also affected by temperature [80]. In the presence of diesel, temperature can influence the active bacterial community structure because only some species are responsible for the biodegradation of specific hydrocarbons [81,82]. Alkanes are the first hydrocarbons to be utilised [83], as they are more bioavailable. The resulting proportional increase in aromatic hydrocarbons remaining will generate a dynamic shift within the bacterial consortium, with an increase in the fraction of bacteria that can utilise these more complex compounds [84,85].

#### 4.1.4. Nitrogen: Source and Concentration

The enhancement in the degradation achieved in the presence of NH_3_NO_4_ (Figure 4a) may be explained by the fact that hydrocarbons exist in a reduced state, and they are oxidised by microbes using electron acceptors. Since oxygen can also become a limiting factor, electron acceptors added to substitute for oxygen may indicate that the nitrate ion provided the next best alternative [86]. Nitrate gives high oxidation potential for the removal of hydrocarbon contamination [87]. Moreover, the additional N can act as a macronutrient supporting the synthesis of amino acids and nucleic acids and allowing rapid cell growth in the medium [88]. Diesel reduction was considerably higher in NH_3_NO_4_, urea, NH_4_Cl and NH_4_SO_4_. However, the choice of the nitrogen source is also influenced by cost and environmental impacts. Urea is considered to be an expensive source of nitrogen in comparison to NH_3_NO_4_ [89]. NH_3_NO_4_ is predominantly used and manufactured for applications in agriculture, mining, quarrying and civil construction [90].

At higher concentrations of ammonium nitrate, diesel degradation began to reduce, suggesting more studies are required to properly ascertain the amount of nitrogen needed to sustain the key microbial players needed for diesel degradation. It is important not to indiscriminately add nitrogen because of the potentially harmful impact its overfertilisation will have on the environment. Microbial communities utilise fuel as a source of carbon and increase the demand for inorganic nutrients like nitrogen [91]. As a result, increased nutrient demand makes nitrogen bioavailability a limiting factor that controls hydrocarbon biodegradation in marine oil-impacted environments [92]. The use of nitrogen has also been reported to be optimum at salinity levels of 20–25 ppt, whereas lower nitrogen turnover occurred at salinity 30 ppt [93]. It has been established that N can improve the biodegradation of hydrocarbons of marine microorganisms. Even though inorganic N, such as NO_3_^−^ and NH_4_^+,^ limits the extent of hydrocarbon degradation in the marine environment, optimum concentrations can benefit the process of bioremediation of oil-contaminated marine environments [94].

#### 4.1.5. Diesel Concentration

Some microbial species are known to tolerate high concentrations of diesel [42,95]. The initial substrate concentration affects the uptake and degradation of the hydrocarbon compound [96]. As noted above, when exposed to a mixture of hydrocarbons, as would be the case in a diesel spill, initially the more readily biodegradable components will be degraded, leading to rapid growth and high levels of respiration [97]. Alkanes of intermediate chain length are typically degraded rapidly, whereas longer chain alkanes and the more toxic aromatic compounds are more recalcitrant and persist for a longer period of time [53]. In the current study there was a considerable reduction in diesel biodegradation at the highest initial diesel concentration trialled (5.5% *v/v*; Figure 5), suggesting that the key biodegrading members of consortium o2b could not tolerate that high of a concentration.

### 4.2. Application of Response Surface Methodology in the Selection of Significant Variables

Screening using PBD evaluates the influence of each of the selected environmental parameters separately on the degradation process [98]. In this study, all of the independent environmental factors were significant. The subsequent use of CCD allowed identification of the significant pairwise interactions between them (Figure 6). The optimisation results obtained through RSM revealed that the data fitted well with the quadratic model (Table 7). CCD predicted higher diesel biodegradation at lower salinity and higher nitrogen concentration. However, the response surfaces generated also support the adaptability or flexibility of the hydrocarbon-degrading members of bacterial consortium o2b, which can tolerate varying salinity levels while still maintaining considerable, if sub-optimal, biodegradation ability.

Temperature was a key factor influencing biodegradation of hydrocarbons by bacterial consortium o2b. In general, chemical reactions obey the Arrhenius relationship, where rates increase with temperature, modulated in biological enzyme-mediated reactions by the properties of the enzymes involved. The data obtained here clearly show that consortium o2b achieved maximum hydrocarbon degradation at the relatively low temperature of 10–12 °C (Figure 6c). A further effect of temperature on the rate of hydrocarbon degradation acts through its influence on N availability (Figure 6b). This is consistent with a study in the sub-Antarctic Kerguelen archipelago, where temperatures of 10 °C and 20 °C, in the presence of diesel fuel and varying nitrogen concentrations, induced changes in the total bacterial consortium in seawater, while at 4 °C the consortium structure remained stable [77]. Temperature also affects the availability of the oil due to the physical nature of the hydrocarbons.

Community interactions in natural consortia are of paramount importance when considering the biodegradation of complex hydrocarbon mixtures, such as crude oil and diesel. A natural consortium can degrade a complex mixture of hydrocarbons more effectively because of complementary interdependence between its members [99]. Such interactions may be crucial for the preliminary steps that eventually lead to the mineralisation of the hydrocarbon compounds. Most research to date has focused on hydrocarbon degrading marine microbial taxa, ignoring any nitrifying species present. Nitrifiers, which comprise both bacteria and archaea, are more sensitive to hydrocarbon toxicity than are typical heterotrophs. Thus, when an oil spill occurs in coastal ecosystems, a loss of nitrification activity follows. Denitrification is then driven by heterotrophs, slowing down nitrogen turnover and as a consequence, increasing ammonia availability, which supports the growth of oil-degrading heterotrophs. Nitrifying bacteria naturally present in the environment are also expected to play a role in remediating oil [100]. For ammonium-oxidising bacteria, exposure to hydrocarbons causes disruption to the outer membrane of the cell, along with the formation of toxic products from hydrocarbon metabolism that are cytotoxic [101].

## 5. Conclusions

This study examined the ability of marine bacterial consortia from uncontaminated Antarctic seawater to biodegrade diesel. The optimisation of the consortium o2b through the application of both conventional OFAT and the statistical approach of RSM, yielded a reduction of 12.23 mg/mL from the initial 30.518 mg/mL concentration of diesel (4% *v/v*). The optimisation of diesel biodegradation also provided a better understanding of how different parameters influenced one another in response to diesel biodegradation by an Antarctic bacterial consortium, confirming that salinity, nitrogen concentration, initial diesel concentration and temperature all significantly influenced diesel degradation. Further significant pairwise interactions were found between salinity and nitrogen, nitrogen and temperature and diesel concentration and temperature. Consortium studies provide a realistic assessment of how natural microbial communities are affected by hydrocarbon pollution and of how changing environmental conditions could impinge on different consortia.

## Figures and Tables

**Figure 1 microorganisms-09-01213-f001:**
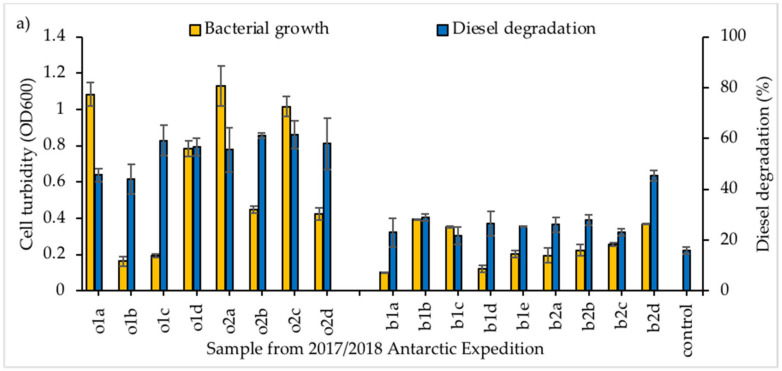
Screening of bacterial consortia growth and diesel degradation obtained from Antarctic seawater samples collected in (**a**) 2017/18 and (**b**) 2018/19; all consortia grown in BH media, supplemented with 20 ppt NaCl and 1.0% initial diesel concentration as sole carbon source. Vertical bars indicate SEM of three replicates.

**Figure 2 microorganisms-09-01213-f002:**
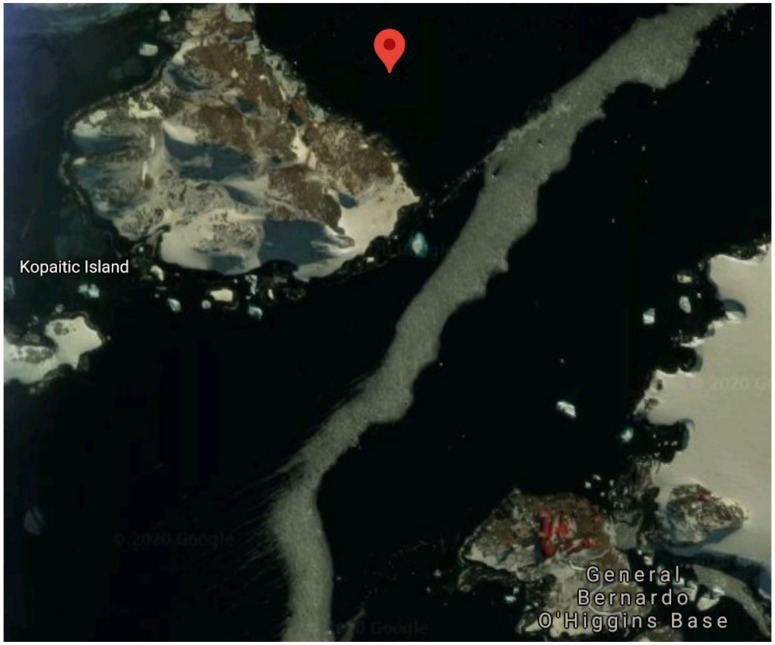
Aerial photograph showing source location of selected consortium sample. (Source: Google.com).

**Figure 3 microorganisms-09-01213-f003:**
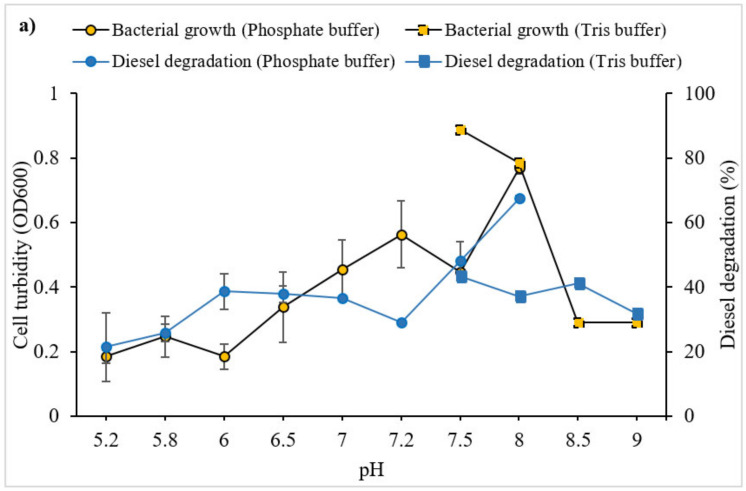
Effect of (**a**) varying pH (using two buffer systems), (**b**) salinity (% *w*/*v*) and (**c**) temperature on bacterial consortium growth and diesel degradation. Vertical bars indicate SEM of three replicates.

**Figure 4 microorganisms-09-01213-f004:**
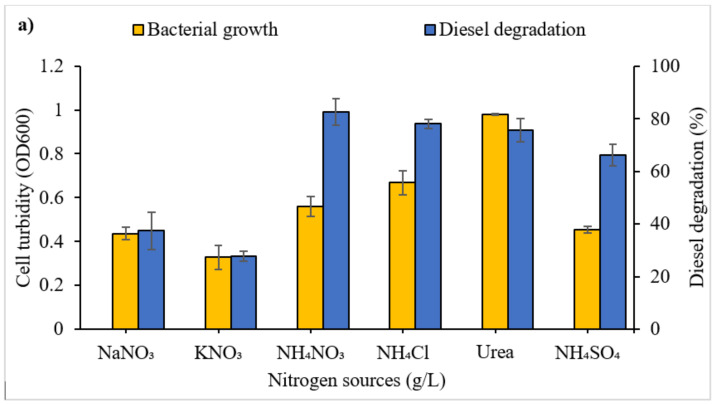
Effects of (**a**) different nitrogen sources and (**b**) of the selected nitrogen source concentration on bacterial consortium growth and diesel degradation. Vertical bars indicate SEM of three replicates.

**Figure 5 microorganisms-09-01213-f005:**
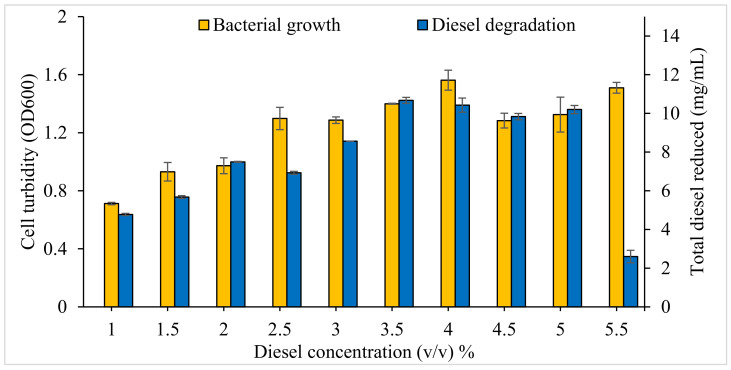
Effects of initial diesel concentration (*v*/*v*) on bacterial consortium o2b growth and diesel degradation. Vertical bars indicate SEM of three replicates.

**Figure 6 microorganisms-09-01213-f006:**
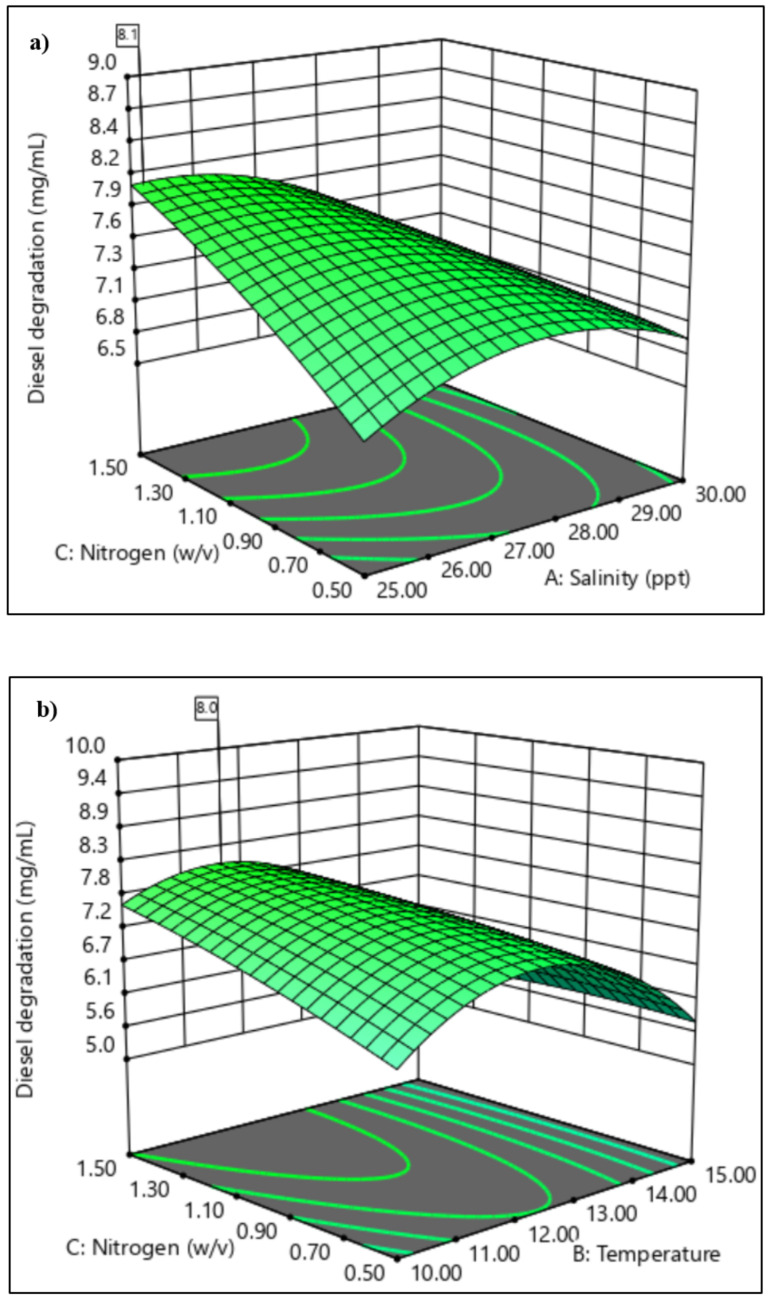
3D Contour plots generated by Design Expert (Stat Ease, Inc) of the significantly interacting model terms (**a**) A: salinity and C: NH_3_NO_4_ concentration, (**b**) B: temperature and C: NH_3_NO_4_ concentration, (**c**) B: temperature and D: diesel concentration.

**Table 1 microorganisms-09-01213-t001:** Parameters used and their ranges used in the OFAT experiment.

Parameter	Range
pH (potassium phosphate and Tris buffer)	5.2–9.0
Salinity, NaCl (ppt, *w/v*)	0–50
Temperature (°C)	10–25
Nitrogen source	urea, NH_4_NO_3,_ NaNO_3,_ KNO_3,_ NH_4_SO_4_
Nitrogen concentration, g/L (*w/v*)	0–3.5
Initial substrate concentration, diesel (%, *v/v*)	0.5–5.0

**Table 2 microorganisms-09-01213-t002:** Experimental values and levels of variables tested for bacterial consortium in Plackett-Burman design.

Variables	Code	Unit	Experimental Range	
			Low (‒1)	High (+1)
pH	A	-	7.5	8.0
Salinity	B	ppt	25.0	30.0
Temperature	C	°C	10.0	15.0
Nitrogen concentration	D	g/L	0.5	2.5
Initial diesel concentration	E	% (*v/v*)	1.0	4.0

**Table 3 microorganisms-09-01213-t003:** Experimental values and levels of the selected independent factors for CCD optimisation.

	Symbol	Unit	Experimental Values				
			‒2	‒1	0	+1	+2
Salinity	A	ppt	22.5	25.0	27.5	30.0	32.5
Temperature	B	°C	7.5	10.0	12.5	15.0	17.5
Nitrogen concentration	C	(g/L)	0.0	0.5	1.0	1.5	2.0
Initial diesel concentration	D	% (*v/v*)	0	1.0	2.5	4.0	5.5

**Table 4 microorganisms-09-01213-t004:** Secondary screening of significant parameters affecting diesel degradation using Plackett-Burman design matrix for bacterial consortium o2b (±SEM, *n* = 3).

Run	A	B	C	D	E	Degradation (mg/mL)
1	8.0	30.0	15.0	0.5	30.51	9.62 ± 1.54
2	7.5	25.0	15.0	2.5	30.51	5.76 ± 2.51
3	8.0	30.0	10.0	2.5	30.51	5.17 ± 0.37
4	7.5	25.0	10.0	2.5	30.51	5.06 ± 1.06
5	7.5	30.0	15.0	2.5	7.69	2.34 ± 2.24
6	8.0	30.0	10.0	2.5	7.69	2.05 ± 1.01
7	8.0	25.0	15.0	2.5	7.69	2.10 ± 1.99
8	8.0	25.0	10.0	0.5	30.51	8.03 ± 1.98
9	7.5	30.0	10.0	0.5	7.69	2.85 ± 8.17
10	8.0	25.0	15.0	0.5	7.69	3.13 ± 1.37
11	7.5	25.0	10.0	0.5	7.69	2.89 ± 2.51
12	7.5	30.0	15.0	0.5	30.51	9.71 ± 1.02

A: pH; B: Salinity (ppt); C: Temperature (°C); D: NH_3_NO_4_ concentration (g/L); E: Initial diesel concentration (mg/mL).

**Table 5 microorganisms-09-01213-t005:** ANOVA of the PBD model used to identify the factors significantly influencing diesel biodegradation.

Source	Sum of Squares	DF	Mean Square	F Value	*p* Value
Model	629.47	5	125.89	164.52	<0.0001 ***
A	0.5663	1	0.56	0.74	0.4227
B	4.62	1	4.62	6.04	0.0493 *
C	32.12	1	32.12	41.99	0.0006 ***
D	376.17	1	376.17	491.59	<0.0001 ***
E	215.98	1	215.98	282.24	<0.0001 ***
Residual	4.59	6	0.7652		
Cor Total	634.06	11			
Std. Dev.	0.87.48	R^2^		0.9928	
Mean	27.92	Adjusted R^2^		0.9867	
C.V.	3.13	Predicted R^2^		0.9710	
		Adequate Precision		36.4091	

A: pH; B: Salinity (ppt); C: Temperature (°C); D: NH_3_NO_4_ concentration (g/L); E: Initial diesel concentration (mg/mL); * *p* < 0.05, ** *p* < 0.01, *** *p* < 0.001.

**Table 6 microorganisms-09-01213-t006:** Optimisation of parameters for diesel degradation by bacterial consortium o2b using central composite design (CCD) (±SEM, *n* = 3).

Run Order	A	B	C	D	Diesel Reduction (mg/mL)	
					Experimental Value	Predicted Value
1	27.5	12.5	1.0	44.37	15.01 ± 3.54	14.84
2	25.0	15.0	0.5	7.96	3.15 ± 8.67	2.71
3	30.0	10.0	0.5	7.96	2.61 ± 3.10	2.27
4	25.0	15.0	1.5	30.51	8.31 ± 4.10	8.78
5	27.5	12.5	1.0	18.45	7.74 ± 17.04	7.63
6	27.5	17.5	1.0	18.45	1.63 ± 1.52	1.41
7	25.0	10.0	1.5	30.51	12.39 ± 2.37	11.67
8	25.0	15.0	0.5	30.51	7.66 ± 1.77	7.91
9	30.0	10.0	0.5	30.51	9.49 ± 5.67	9.67
10	27.5	12.5	1.0	0	0.00	0.00
11	32.5	12.5	1.0	18.45	5.83 ± 0.21	5.57
12	30.0	15.0	0.5	7.96	1.89 ± 1.28	2.63
13	30.0	10.0	1.5	7.96	2.88 ± 1.57	2.64
14	27.5	12.5	0.0	18.45	6.43 ± 9.94	6.59
15	27.5	12.5	1.0	18.45	7.88 ± 0.96	7.63
16	25.0	10.0	1.5	7.96	3.87 ± 4.40	4.51
17	25.0	15.0	1.5	7.96	3.11 ± 2.14	2.95
18	30.0	15.0	0.5	30.51	9.2 ± 2.06	8.72
19	27.5	12.5	1.0	18.45	7.72 ± 0.43	7.63
20	25.0	10.0	0.5	30.51	9.16 ± 9.50	9.23
21	30.0	15.0	1.5	30.51	8.08 ± 3.67	8.21
22	27.5	12.5	1.0	18.45	7.85 ± 0.04	7.63
23	27.5	12.5	1.0	18.45	7.09 ± 0.99	7.63
24	30.0	10.0	1.5	30.51	10.09 ± 3.33	10.69
25	27.5	12.5	2.0	18.45	8.18 ± 2.75	7.86
26	27.5	7.5	1.0	18.45	3.86 ± 1.02	3.91
27	27.5	12.5	1.0	18.45	7.48 ± 13.19	7.63
28	30.0	15.0	1.5	7.96	1.37 ± 17.36	1.46
29	22.5	12.5	1.0	18.45	6.54 ± 2.76	6.62
30	25.0	10.0	0.5	7.96	2.84 ± 6.79	2.73

A: Salinity (ppt); B: Temperature (°C); C: NH_3_NO_4_ concentration (g/L); D: Initial diesel concentration (mg/mL).

**Table 7 microorganisms-09-01213-t007:** Results of ANOVA for CCD model identifying factors and pairwise interactions significantly influencing diesel biodegradation.

Source	Sum of	Df	Mean	F-Value	*p* Value	
Model	307.675	14	21.977	85.488	<0.0001	***
A	1.649	1	1.649	6.415	0.0239	*
B	9.397	1	9.397	36.553	<0.0001	***
C	2.411	1	2.411	9.377	0.0084	**
D	188.075	1	188.075	731.598	<0.0001	***
AB	0.142	1	0.142	0.552	0.469	
AC	1.987	1	1.987	7.728	0.014	*
AD	0.791	1	0.791	3.078	0.101	
BC	2.369	1	2.369	9.213	0.009	**
BD	1.707	1	1.707	6.642	0.022	**
CD	0.432	1	0.432	1.681	0.216	
A^2^	3.907	1	3.907	15.197	0.002	**
B^2^	41.091	1	41.091	159.839	<0.0001	***
C^2^	0.273	1	0.273	1.060	0.321	
D^2^	0.347	1	0.347	1.351	0.265	
Residual	3.599	14	0.257			
Lack of Fit	3.149	9	0.350	3.883	0.075	Not significant
Pure Error	0.450	5	0.090			
Cor Total	311.274	28				
		R^2^			0.9894	
Std. Dev.	0.507		Adjusted R^2^		0.9769	
Mean	6.53		Predicted R^2^		0.9326	
C.V. %	7.76		Adeq Precision		38.8315	

A: Salinity, B: Temperature, C: NH_3_NO_4_ concentration, D: Initial diesel concentration, * *p* < 0.05, ** *p* < 0.01, *** *p* < 0.001.

**Table 8 microorganisms-09-01213-t008:** Model validation using the predicted optima values.

Optimised Parameters	Value	Predicted Value	Experimental Value
pH	8.0	11.66 mg/mL	12.23 mg/mL ± 1.46
Salinity (NaCl)	25.0 ppt		
Temperature	10 °C		
NH_3_NO_4_ concentration	1.5 g/L		
Initial diesel concentration	4.0%		

## Data Availability

Not applicable.

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
