# Peer review of "Statistical Optimisation of Diesel Biodegradation at Low Temperatures by an Antarctic Marine Bacterial Consortium Isolated from Non-Contaminated Seawater"

_microorganisms, 2021, doi:10.3390/microorganisms9061213_

Round 1

Reviewer 1 Report

My comments are presented in the enclosed pdf file.

Author Response

Comment 1

The first question is about employed microorganisms – what types of bacteria it was? Was any identification prepared, to make some further replication of your experiments possible? I found only in manuscript names “o1a, o2a, o2b, o2c” but no explanations. 

Answer: The bacteria samples are labeled as such according to the coordinates and the year it was sampled in. We have included the coordinates in a supplementary document (Supplementary data). As to the question of identification. The authors share your concerns. We are proceeding with a metagenomics approach as a way of identifying the chosen bacteria o2b. Work is still underway but is slightly hampered due to the pandemic as the lab we are relying on is outsourced. Taxonomic identification through metagenomic approach will be carried out as an extension to the current study and the results are expected to be published in different publication.

Comment 2

Figure 1 – there are symbols wb1-wb16, w1-w11 without any description in the text. 

Answer: The coordinates have been included in a supplementary document (Supplementary data).

Comment 3

Figure 4b – in previous plots secondary Y-axis was named as “diesel degradation” but here is “diesel weight reduction” – is that a different parameter? Provide some comments, please. 

Answer: Corrected. Figure 4b has been changed to diesel degradation to follow the previous figures.

Comment 4

CCD presented in Table 3 contains some errors in my opinion. You used five levels of each factor ranging from -2 to 2 (normalized values). However, for initial diesel concentration (D) this creates a very strange situation – you are moving every 1.5% from center point 2.5%, so -2 normalized point is equal to -0.5% as is stated in the table. How could you use a negative initial concentration? I would suggest taking instead the inscribed CCD that does not extents the real ranges, e.g. 0.5 to 4.5%. Then, the star points are located at limits, and the design is scaled down and each factor lever is divided by the alpha value. This comment is related also to other factors (A-C) but there was no critical error like negative concentration. 

Answer: Software Design Expert generated alpha -2 as noise. We have corrected the value to 0. (Table 3).

Comment 5

Reading your manuscript I did not find where is the precise optimum for your experiments. Using RES you obtained a well-fitted model equation and you examined which parameter is significant and which can be neglected. Now you should find the extrema of this function (of four variables) using a mathematical approach. I know that finding the extrema of multi-variable function is not easy, but will give you a precise answer – you can find some software useful to calculate this, e.g. Matlab. Knowing the extrema point you should also make the additional experiment at this point to validate the whole model and see if you got the highest biodegradation of diesel. 

Asnwer: The validation has been carried out based on the optima points from the mathematical model generated by the software. It can be found on Page 14, Line 366-367. The validation for the model can be found on page 14 between Lines 369 – 376. Table 8: ‘Model validation using the predicted optima values’ has been added.

Reviewer 2 Report

The paper describes optimizing conditions for natural Antarctic marine microbial consortia for degrading diesel oil in relevant conditions. Very well written, and mostly clear. A few comments are added to the file. What is presented is good, but the more interesting part would be the consortia themselves. They (or at least the chosen one) hopefully will also be determined and published. Nevertheless, the paper deserves to be published with minor corrections.

Author Response

Comment 1

Section 2.1 “..immediately frozen at -80oC until required.” Were also the samples intended for microbial cultivation frozen? That should be mentioned since it may influence the survival of some strains.

Answer: Yes, the microbial cultivation is also frozen at -80oC because the experiment was done in Malaysia upon arrival. There was only a few weeks in Antarctica and -80oC was the only equipment of preservation that was available at the laboratory in Antarctica.

Comment 2

Table 2: pH 7.5 (Low -1). Explain the some point the narrow range of pH. I guess the ranges tested in total were broader, but a more narrow range was used in the PBD. I may be explained somewhere, but I have not seen it yet.

Answer: Initially the tested ranges were broader. This experiment was done in the OFAT (Figure 3a). The optimal range was shortened to pH 7.5 – 8.0 before proceeding with the PBD.

Comment 3

Table 3: -0.5. What does -0.5 mean?

Answer: Value -0.5 here refers to the initial concentration of diesel. The value is generated by the software and experimentally it was placed at 0% diesel. This run generated an outlier and was thus excluded from the ANOVA. The value has been corrected; the value is 0. (Table 3).

Comment 4

4. Discussion. “microbial communities” I guess the actual communities have also been determined but published elsewhere. For me that is the interesting part/.

Answer: Thank you for your comment. The authors share your concerns. We are proceeding with a metagenomics approach as a way of identifying the chosen bacteria o2b. Work is still underway but is slightly hampered due to the pandemic as the lab we are relying on is outsourced. Taxonomic identification through metagenomic approach will be carried out as an extension to the current study and the results are expected to be published in different publication.

Comment 5

4.1.4: “NH3NO4 is predominantly…..” Because of miss-use as explosive access may be restricted.

Answer: Agree. However, with this knowledge, there are general precautions that is taken during the movement and stability whilst handling of these chemicals e.g. storage of up to 50 kg per bag of ammonium nitrate. On a side note, ammonium-based fertilisers have a NPK ratio of 34:0:0 in comparison to another source of NPK such as urea (47.0.0). The experiment requires highest 1.5 g/L of ammonium nitrate which is sufficient for the bacteria to maintain agreeable growth and degrading abilities. The abuse of ammonium nitrate is duly-noted but perhaps it can be handled with caution at the optimum amount only. 

Round 2

Reviewer 1 Report

Thank you for providing your answers.

Author Response

Dear reviewer,

All the questions have been answered in the Attachment file. Thank you.
